# Optimizing Ghanaian Postgraduate students' job performance: The impact of polychronicity, work-school facilitation, and organisational support

Salomey Ofori Appiah[1]*, Richard Kofi Boateng[2]

1 Department of Human Resource Management, University of Cape Coast, Ghana, 2 Department of Human Resource Management, University of Cape Coast, Ghana

* salomey.ofori@ucc.edu.gh

## Abstract

This study examined the influence of polychronicity, work-school facilitation, and organizational support on the job performance of postgraduate students in Ghana. Guided by the Conservation of Resources (COR) theory, the research explored how personal and organizational resources affected job performance outcomes. A quantitative, explanatory research design was used. A sample of 341 distance education postgraduate students from three Ghanaian universities was selected using convenience sampling. Data were collected via a three-time lagged survey method to minimize common method bias. Structural Equation Modeling (SEM) using SMART PLS software (version 3.98) was employed for data analysis. The findings revealed that polychronicity had a significant positive effect on job performance. Organizational support moderated this relationship, strengthening the effect of polychronicity on job performance. Additionally, work-school facilitation mediated the relationship between polychronicity and job performance, suggesting that students who effectively balanced academic and work roles performed better. Polychronicity also directly influenced work-school facilitation. The study provided both theoretical and practical insights, highlighting the importance of individual time-use preferences and institutional support in enhancing the performance and well-being of postgraduate students.

## Introduction

In today's fast-paced and dynamic environment, efficiently juggling multiple tasks and responsibilities is not just an advantage but a necessity [1]. This is particularly true for postgraduate students, who often face the dual demands of rigorous academic pursuits and challenging professional roles. Balancing these priorities is a complex challenge, reflecting the broader struggles of modern education and work-life balance, especially in developing economies [2]. Statistics indicate that over 60% of master's students and nearly 50% of doctoral students in the United States work part-time or

**Data availability statement:** All relevant data are within the manuscript and its Supporting Information files.

**Funding:** The author(s) received no specific funding for this work.

**Competing interests:** NO The authors have declared that no competing interests exist.

full-time, with many experiencing heightened stress, academic fatigue, and burnout compared to their non-working peers [3,4]. This dual commitment often results in decreased productivity, lower grades, and extended completion times for degrees [5]. Similar trends exist in sub-Saharan Africa, where cultural values and orientations pose additional challenges for individuals balancing multiple responsibilities [6]. For instance, many Ghanaian students prefer to focus on their studies before engaging in active work to avoid the stress and counterproductive behaviors associated with juggling both [7]. The pressure of meeting work and academic deadlines often compromises the quality of coursework and research, while professionally, students may exhibit reduced concentration, lower efficiency, and increased absenteeism [8].

To navigate these challenges, the Conservation of Resources (COR) theory posits that individuals are motivated to acquire, retain, and protect resources that help them cope with stress [9]. Building on this, [10] argue that polychronicity, the ability to manage and excel in multiple tasks simultaneously acts as a valuable personal resource for addressing the competing demands of work and study. However, research on the relationship between polychronicity and job performance has yielded mixed results. Some studies highlight its positive impact on performance [11–13] found that polychronicity has a significant effect on job performance, while others report an insignificant relationship [14,15]. These conflicting findings suggest that the effect of polychronicity may depend on individual or contextual factors, highlighting the need for further exploration in specific settings, such as Ghana.

Even though personal trait like polychronicity can assist postgraduate students in multitasking, external resources like work-school facilitation can complement this internal resource to provide a supportive structure that allows individuals to effectively manage their roles as both employees and students. Work-school facilitation encompasses strategies, practices, and support systems designed to help individuals successfully manage their dual roles as employees and students [16]. This facilitation aims to create a synergistic relationship between work and education, where each role positively influences the other, enhancing personal development, career advancement, and academic achievement [17].

According to [18], people employ various strategies to integrate or segment their roles and responsibilities. In this context, work-school facilitation allows individuals to acquire skills in the workplace, such as time management, problem-solving, and teamwork, which can be beneficial in an academic setting. Conversely, analytical skills, research methods, and critical thinking learned at school can enhance the job performance of these students. Although this proposition may be true, limited research has been conducted to probe into how polychronicity affects work-school facilitation. Again, studies on how work-school facilitation translates polychronicity into superior performance have received a muted voice in the academic literature.

While work-school facilitation can be a lens through which the performance of postgraduates with polychronic orientation can be enhanced, the study of [10] recommended that researchers who are interested in understanding the polychronic orientation of individuals and how it affects workplace outcomes should also consider the role of organisational support. Indeed, organisational support which reveals the

extent to which supervisors and coworkers care and provide support for employees has been found as a medium through which positive individual attitudes translate into performance. For instance, providing resources to employees improves their job performance and reduces burnout [19]. Additionally, it has been found that employees who perceive support from their organization and coworkers can increase their performance by 15% thereby increasing their institution's chances of competing with other firms [20–23]. Similarly, [24] emphasizes that employees can leverage adequate resources, such as support, to enhance performance. As such, employees can use those resources to execute the demands of the job.

On this premise, the study contends that when postgraduate students have access to adequate support from their respective organizations, they can withstand the pressures arising from multitasking tasks from school and work duties which leads to an increase in job performance. However, this claim has not been documented in the literature, and hence this study would fill this void.

This study is premised on four main objectives. First, the study intends to address the effect of polychronicity on job performance. Secondly, this study would provide an idea of how work-school facilitation translates polychronicity into job performance, as this relation has received a muted voice in the literature. Finally, the study would provide insights into how organisational support moderates the relationship between polychronicity and job performance. By probing into these objectives, the study's contributions can inform strategies for educational institutions and employers to better support postgraduate students, potentially leading to improved academic outcomes, enhanced job performance, and a healthier work-life balance. Theoretically, this research enriches the conversation around multitasking, resource management, and the dynamics of balancing multiple high-stakes roles, providing a foundation for future studies to build upon.

## Theoretical and hypotheses development

The Conservation of Resources (COR) theory was developed by [9]. The COR theory posits that individuals are driven to acquire, protect, and retain valued resources both organisational and personal. That is employees at any point in time would seek to obtain new resources that can assist them in coping with stressful situations. In the context of this study, organisations that help their employees in resource management and provide support systems can help improve the job performance of individuals (postgraduate students). For instance, individuals with high polychronicity potentially maintain and juggle various resources such as time, attention, and energy more effectively. This multitasking ability can be considered a vital resource itself, which reduces the stress associated with balancing multiple responsibilities and enhances overall performance and success. Research indicates that individuals with high polychronicity tend to adapt better in dynamic environments, potentially enhancing their performance [25]. This adaptability can be seen as a resource gain, as efficient multitasking reduces stress and enhances overall performance, aligning with COR theory's principles.

Again, the ability of individuals to facilitate work and school (work-school facilitation) activities is a resource that reinforces positive spillover where skills, behaviors, and attitudes from one role enhance the quality of life in another [26]. For postgraduate students, the ability to transfer skills and knowledge between work and academic settings can transfer skills learnt in school to improve their efficiency in their work. The COR theory suggests that these new resources, whether they are new skills or expanded networks, can act as buffers against potential stressors in both domains, thereby promoting better work outcomes and positive employee behaviour. Effective facilitation allows skills and experiences from one domain to benefit the other, creating a resource gain that reduces stress and enhances performance. Within the COR framework, work-school facilitation enables the transfer of resources between work and academic domains, reducing stress and enhancing performance.

Finally, organisational support serves as a critical resource in this relationship. Perceived organisational support leads to higher job satisfaction and better job performance among employees [26]. In the context of postgraduate students in Ghana, where balancing work and academic responsibilities is common, organisational support provides students with the necessary resources and assistance to manage their multiple roles effectively. This support reduces stress and enhances job performance. The theory pinpoints that support from academic workplaces, such as flexible schedules or emotional

support from colleagues and supervisors, directly contribute to students' resource pools. This organisational support helps mitigate the demands placed on students, thereby preventing resource depletion. That is, the COR theory pinpoints that shoring up resources such as support ensures that students are less vulnerable to stressors that might otherwise impede their performance in their organisation. These relationships have been presented in Fig 1.

## Moderation of organisational support on the effect of polychronicity on job performance

Polychronicity describes an individual's ability for multitasking and managing several activities at once [27]. Polychronicity, an individual's capacity to multitask and manage several activities simultaneously, is a critical trait in dynamic work environments. Individuals high in polychronicity thrive on flexibility and adaptability, which are essential in fast-paced and unpredictable settings. Studies by [27] and [28] indicate that these individuals are adept at switching between tasks, managing interruptions, and handling overlapping responsibilities. This ability fosters creativity, enhances problem-solving, and contributes to superior job performance, particularly in roles that demand responsiveness and multitasking.

From a theoretical perspective, Conservation of Resources (COR) Theory [29] explains how multitasking abilities enhance cognitive flexibility, enabling individuals to adapt and innovate effectively. Polychronic individuals leverage this flexibility to manage multiple roles and responsibilities. In organizational contexts, this adaptability can enhance team collaboration, increase efficiency, and ultimately improve group and organizational performance. [30] support these claims, highlighting that polychronic individuals perform better in environments that require creativity and responsiveness. However, the effectiveness of polychronicity is influenced by specific boundary conditions. One key factor is the availability of organizational support, such as flexible work structures, access to resources, and understanding management. The Conservation of Resources (COR) Theory [9] suggests that while high polychronicity increases the need for supportive structures, the absence of such support can lead to stress and burnout. [14] and [15] found that in less supportive or rigid work environments, the relationship between polychronicity and job performance becomes insignificant. Without adequate resources or flexibility, polychronic individuals may struggle to manage their responsibilities effectively, resulting in decreased performance and heightened stress.

Organizational support also plays a crucial role in moderating the impact of polychronicity on job performance. A positive organizational culture that fosters teamwork, communication, and innovation provides a conducive environment for multitasking individuals to thrive. According to Social Exchange Theory [31], supportive workplace cultures encourage reciprocal behaviors, motivating employees to leverage their multitasking abilities for the benefit of the organization. [21] emphasize that such cultures allow polychronic individuals to collaborate with colleagues, adapt to changing

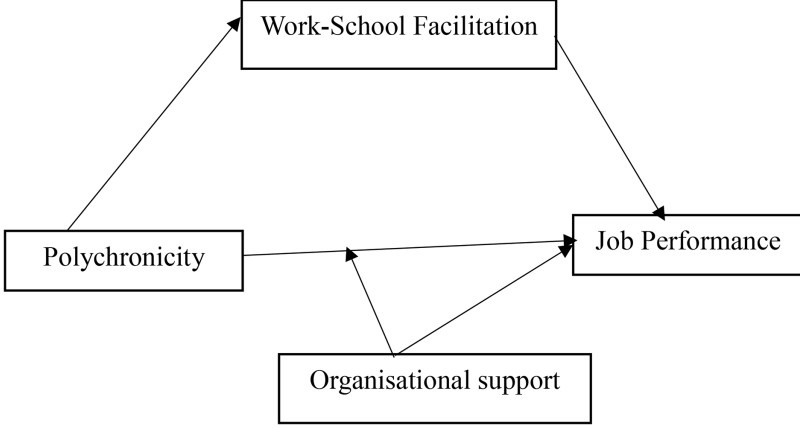

**Fig 1. Conceptual Framework.**

circumstances, and contribute to organizational success. In contrast, when organizational support and culture are lacking, the potential benefits of polychronicity may be undermined. Again [32] highlights that rigid work settings, a lack of professional development opportunities, and limited access to resources can diminish the advantages of multitasking. This creates an environment where the demands of multitasking become overwhelming, leading to role conflict, stress, and reduced performance.

Empirical evidence highlights the importance of aligning organizational support with the needs of polychronic individuals. For instance, [11] and [12] found that multitasking abilities positively influence job performance in flexible and resource-rich environments. Conversely, studies by [14] and [33] affirm that even high polychronicity may not yield better performance in the absence of supportive structures. This dichotomy highlights the critical role of organizational design in maximizing the potential of polychronic individuals. The COR theory highlights that high polychronicity may increase the need for organisational support to manage multiple tasks efficiently, thereby moderating its impact on job performance [27].

Based on these discussions, this study proposes that;

*H1; polychronicity has a significant positive effect on job performance.*

*H2; organisational support significantly moderates the positive effect of polychronicity on job performance.*

## Mediation of work-school facilitation on the effect of polychronicity on job performance

Work-School Facilitation describes the positive interplay between academic and professional domains, where skills and strategies developed in one area are transferred and applied to the other [34,35]. This dynamic relationship equips individuals with time management, organizational, and problem-solving skills, as highlighted by [36]. These skills, cultivated while balancing academic and work demands, enhance employees' efficiency and effectiveness in their job roles. For instance, students who learn to prioritize study sessions and manage work projects simultaneously are better equipped to meet workplace deadlines and handle overlapping responsibilities. This ability to transfer and apply skills across domains underscores the practical benefits of Work-School Facilitation in professional settings.

Empirical studies further validate the impact of Work-School Facilitation on employee performance. For instance, [33,37] and [38] found that the balancing act required to juggle work and school responsibilities fosters resilience and adaptability. These traits are critical for navigating the challenges of dynamic and unpredictable job environments. Polychronic individuals, in particular, thrive in such contexts, as their experience in switching between academic and professional tasks enhances their multitasking abilities. [39] emphasized that this adaptability enables such individuals to manage work and school pressures effectively, respond to sudden changes, and perform multiple tasks seamlessly, ultimately contributing to improved job performance.

Work-School Facilitation extends beyond skill enhancement, contributing significantly to personal growth and career development. The intellectual stimulation gained from academic pursuits fosters continuous learning, enhancing creativity, cognitive abilities, and critical thinking skills [40]. These enhanced capabilities translate into innovative problem-solving and more effective decision-making in the workplace. For example, a student-employee exposed to advanced concepts in their field of study can leverage this knowledge to improve work practices, produce higher-quality outputs, and demonstrate greater job performance [22].

The positive effects of Work-School Facilitation are also evident in its influence on long-term career development. The resilience, adaptability, and enhanced cognitive skills gained through this dual engagement empower individuals to navigate career challenges and seize professional opportunities more effectively. Furthermore, the continuous feedback loop between academic learning and workplace application fosters a growth mindset, enabling employees to adapt to new roles, innovate, and excel in their careers.

Hence the study hypothesis that-

*H3; polychronicity has a significant positive effect on work-school facilitation*

*H4; Work School facilitation has a significant positive effect on job performance.*

*H5; Work School facilitation significantly mediates the positive effect of polychronicity on job performance.*

The proposed conceptual model integrates polychronicity, organizational support, Work-School Facilitation, and job performance, underpinned by empirical evidence. Polychronicity, the ability to multitask and manage overlapping demands, directly enhances job performance by fostering adaptability and responsiveness (Korabik et al., 2017; Howard & Cogswell, 2023). However, its effectiveness is contingent on organizational support, which moderates this relationship by providing essential resources and a flexible work environment. Studies show that in supportive settings, polychronicity leads to higher performance, while its absence can result in stress and diminished outcomes (Karatepe et al., 2013; Wu et al., 2020).

Additionally, Work-School Facilitation mediates the relationship between polychronicity and job performance, enabling the transfer of skills such as time management and problem-solving from academic to professional domains (Mattarelli et al., 2015; Nicklin et al., 2019). This facilitation fosters resilience, adaptability, and creativity, which are critical for navigating dynamic job environments (Koperski, 2017; Bertolotti et al., 2019). The intellectual stimulation from academic engagement further enhances cognitive flexibility and innovative problem-solving, translating into superior workplace performance (Ridwan et al., 2020; Kerse & Çil, 2024). By integrating individual traits, supportive environments, and cross-domain learning, the model provides a comprehensive framework for understanding how polychronicity interacts with organizational and developmental factors to drive job performance.

## Methodology

This study examined the impact of Polychronicity, Work-school Facilitation, and Organisational Support on Ghanaian Postgraduate students' success. This study used the quantitative approach and explanatory study design. The population for this study was 3000 postgraduate distance education students in three universities in Ghana. More specifically, a sample size of 341 students above 21 years were chosen based on the Krejecie and Morgan, (1970) sample size determination at a confidence level of 95%. Krejcie and Morgan's formula is suitable for this study as it ensures a statistically reliable sample size for a finite population, maintaining a 95% confidence level with a 5% margin of error. It enhances generalizability by capturing variations within the post-graduate students while balancing practicality and ethical considerations. The formula is widely accepted in social science research as it helps prevent unnecessary over- or under-sampling [41]. Thus, the chosen sample size of 341 is appropriate for drawing valid conclusions. The convenience sample technique was used to select students for the study. While convenience sampling introduces some potential bias, it is often impractical to use randomized sampling in postgraduate research due to accessibility challenges and time constraints [42]. As a result, this study utilized convenience sampling to accommodate students' varied schedules, ensuring participation from those who were available and willing. To address concerns about bias, participants were selected from diverse academic backgrounds. [42] argues that convenience sampling is acceptable in PLS-SEM because it emphasizes predictive accuracy. Additionally, bootstrapping was applied to ensure robust parameter estimation, further mitigating potential biases and enhancing the reliability of the findings. The three universities were chosen because they are the only universities in the country that run distance programs for students who combine work with school on weekends.

Focusing on postgraduate students who are enrolled in distance education programs and also working is particularly significant because these students face unique challenges that differentiate them from traditional, full-time students. Thus, their ability to manage multiple responsibilities and simultaneously balance work, education, and often personal commitments can significantly influence their success in their work performance. This research can provide insights into the

support systems and adaptations necessary to enhance their learning experiences and outcomes. This can lead to more targeted interventions by educational institutions and policymakers to support this growing demographic of students. To ensure anonymity, the Qualtrics method was employed to collect the data. Two field assistants were recruited and trained on (24th February, 2024–10th March, 2024). Afterwards, data was collected over 2 months (15th March, 2024–16th June, 2024).

The data were collected utilizing a three-wave survey approach with a week time interval between surveys [43–45]. This approach is particularly valuable in understanding changes in attitudes, behaviors, or conditions over time, offering insights into trends, causal relationships, and the dynamics of change [46]. Each "wave" represents one round of data collection, and by comparing data across these three periods, researchers can identify patterns of change, stability, or regression among the participants. The multi-wave design was selected to overcome common method bias due to the challenge of collecting the data for all tested variables at the same point in time [45,24]. The three surveys encompassed Survey 1 for the independent variables and screening questions to ensure that the respondents met the sample require-ments. Survey 2 consisted of the scale for the moderating variable. The data for the dependent variable and the demo-graphic information were collected in Survey 3. However, due to non-response rate 370 questionnaires were distributed. The final sample after data cleaning consisted of 341 employees, which consisted of 56.5% males and 43.5% females.

## Ethics statement

All participants provided written informed consent for the collection of information and the publication of data generated by the study. The target population for this study was postgraduate distance students who were over 21 years and were deemed as adults per the constitution of Ghana. All participants voluntarily partake in the survey and may withdraw at any time. Consid-ering that the survey content does not involve any sensitive issues and that the data collected are completely anonymous, according to the standards of the colleges of distance education in Ghana, this study does not require ethical review, however, an introductory letter was obtained. Furthermore, the research does not pose a risk to the physical or psychological health of participants, and the content of the survey was conducted in a manner that respects and protects the privacy of participants.

## Measurement

The study variables were measured based on empirical studies. Specifically, validated items were adapted from previous studies. In this view, 5-items for polychronicity were adapted from the empirical work of [47]. Again, 5-items for work-school facilitation and -items for organisational support were adapted from [48,49] respectively. And finally, 5-items for job performance were adapted from [50]. All items were measured on a 5-item Likert scale ranging from least agreement to high agreement. That is all measures are anchored on a five-point Likert-type scale ranging from 1 ("least agreement") to 5 ("strongly agree").

To ensure that all respondents participated in the study, we collected the data at different intervals to take care of the differ-ent free time available for the respondents. In this regard, part of the data was collected in the morning, while the rest was also collected in the afternoon. The structural equation modeling software was used in the analysis. In this regard, the PLS model was used to assess the quality criteria of the constructs used. As ethics demands, the respondents were also informed of their role in providing valued information and the purpose for which the information is going to be used. The respondents were fur-ther given assurance of anonymity and confidentiality and were also informed of the voluntary nature of the survey. To enforce confidentiality, anonymity, and privacy, the questionnaire content did not request personal identification.

## Data analysis

The data's accuracy was double-checked following a careful assessment. Following the coding of the question-naires, the data was entered into the Statistical Package for Social Sciences (SPSS) version 25. SPSS was used to organize and summarize data, as well as to provide critical parameters for data analysis. The data was analyzed

using the SMART PLS software (version 3.98). The same applicability is acknowledged for investigations of this sort [51,52]. The reflective structural model was configured using the second-order method. Model setup criteria were developed using recommendations from previous studies [51,53,54]. The configured model was evaluated using the suggestions of [51]. The two-stage approach to structural model evaluation was strictly followed. During the model evaluation step, non-significant indicators with less than a 0.7 factor score were eliminated. This is based on the assumption that their removal enhances the quality dimensions of the configured model. Items with a factor score of less than 0.7 are retained if their removal negatively influences the quality dimensions of the configured structural model [55].

Additionally, this study carried out a common method bias (CMB) test to ensure that the research data were not contaminated with data bias. This is pertinent in cross-sectional research, particularly those that adapt self-report questionnaires such as the present study. The difference between the genuine correlation among constructs and the observed relationship determined using common method variance (CMV) is measured with the CMB [56,57]. The disparity between perceived and actual correlations, facilitated by CMV, poses a significant threat to the validity and reliability of the research findings [58]. As a result, [59] as cited by [57] suggested a combination of procedural and statistical improvements for controlling and minimizing CMV. In this study, we adopted two methods. Procedurally, we used the time lag to collect data at different periods and statistically, we used the correlation matrix method and the Harman single-factor test. [60] and [61] as cited by [57] signpost that the coefficient between constructs should be less than 0.90 and the Harman single-factor test should be less than 50%. The attainment of these thresholds proves the absence of common method bias. The results of the correlations among the constructs and the Harman single-factor test are present in Table 1 and Table 2 respectively.

## Results

### Mean, standard deviation and correlation coefficient of variables

Table 1 shows the results of correlation analysis between the study variables. The study found a significant positive correlation between polychronicty and job performance (r = 0.473, p < 0.001), organizational support and job performance (r = 0.663, p < 0.001), and polychronicty and Work School Facilitation (r = 0.405, p < 0.001), which provides preliminary support for the positive prediction of family economic stress on career exploration. These results not only validate the rationality of our hypothesis but also provide important data support for the subsequent hypothesis testing.

**Table 1. Descriptives and Correlation Analysis.**

|  | Mean | Standard Deviation | 1 | 2 | 3 | 4 |
|---|---|---|---|---|---|---|
| 1. Job Performance | 3.438 | .7329 | 1 |  |  |  |
| 2. Org. Support | 3.175 | .9480 | 0.663** | 1 |  |  |
| 3. Polychronicity | 3.807 | .7338 | 0.473** | 0.313** | 1 |  |
| 4. Work School Facilitation | 3.421 | .9354 | 0.700** | 0.676** | 0.405** | 1 |

Note: **p < 0.01, **p < 0.05 (Same in structural path).

**Table 2. Harman's Single Factor Test.**

|  | Variance Explained by First Factor | Total Variance (Sum of Squared Loadings) | Proportion (%) |
|---|---|---|---|
| Sum of Squared Loadings | 4.3207 | 9.7777 | 44.13 |

## Test for common method bias

The Harman single-factor test was used to check for common method bias. The results showed that the total variance extracted by one factor is 44.13%, less than the suggested threshold of 50% by [56,57,60,61]. Thus, the result shows the absence of CMV (CMB) in the acquired data.

## Measurement model

The study investigated whether the observed variables or indicators could be used to accurately measure the theoretical ideas. However, all of this model's constructs were in response to a reflecting model. In all four constructs Polychronicity (P), Job performance (JP), Organisational support (OS) and Work School Facilitation (WSF) were used. After examining the reliability, an assessment of construct validity followed. The study was able to ascertain the individual dependability of each item in the reliability study by first looking at the factor loadings or straightforward correlations between the indicators and the associated latent variables. If an indicator's factor loading is more than or equal to 0.70, the measurement model for that construct will accept it. This shows that the shared variance of the construct and its indicators is bigger than the variance of the error [62]. However, [63] argue that indicators with loadings between 0.4 and 0.7 should only be eliminated from a scale if doing so results in an increase in the average variance extracted (AVE) or composite reliability (CR). Weak indications may therefore be retained based on how they contribute to the validity of the content. However, it is necessary to constantly delete extremely weak indicators ($\lambda \leq 0.4$). Because they were not increasing the AVE, we deleted indicator loadings that were below the threshold of 0.4 but maintained (OS6) which had a threshold of 0.630. Evidence of the study's reliability and validity check is displayed in Fig 2 and in Tables 2 and 3.

The results in Table 3 showed that the Cronbach Alpha (CA) values for all the items exceeded the minimum threshold of 0.7. These results indicate that the model has internal consistency. Furthermore, Table 3 revealed that the Composite Reliability (CR) for all the constructs was reliable. This is because the constructs had CR scores higher than the 0.7 threshold [64]. Again, the rho_ A results for all the constructs were reliable because they all met the 0.7 minimum criteria. Finally, it was found that AVEs for the constructs accurately measured convergent validity because AVE attained values higher than 0.5. Additionally, the McDonald Omega test was calculated to affirm the reliability of the constructs using:

$$\omega = \frac{(\sum \lambda_i)^2}{(\sum \lambda_i)^2 + \sum \theta_i}$$

Where:
- $\lambda_i$ are the factor loadings,
- $\theta_i = 1 - \lambda_i^2$ are the error variances under standardized assumptions.

The omega values in Table 3 indicate high internal consistency reliability for all constructs, as values above 0.70 are generally considered acceptable, and values above 0.80 are considered good. This showed that the measurement model appears to be well-constructed in terms of reliability.

## Discriminant validity

The results in Table 4 showed that all the constructs accurately measured discriminant validity because all the values were below the 0.9 threshold [51,56].

## Collinearity statistics (VIF)

The results from the collinearity test are presented in Table 4. For the VIF to be acceptable, it must meet a threshold of less than 5 [66]. The Variance Inflation Factor (VIF) was used to measure collinearity in this study.

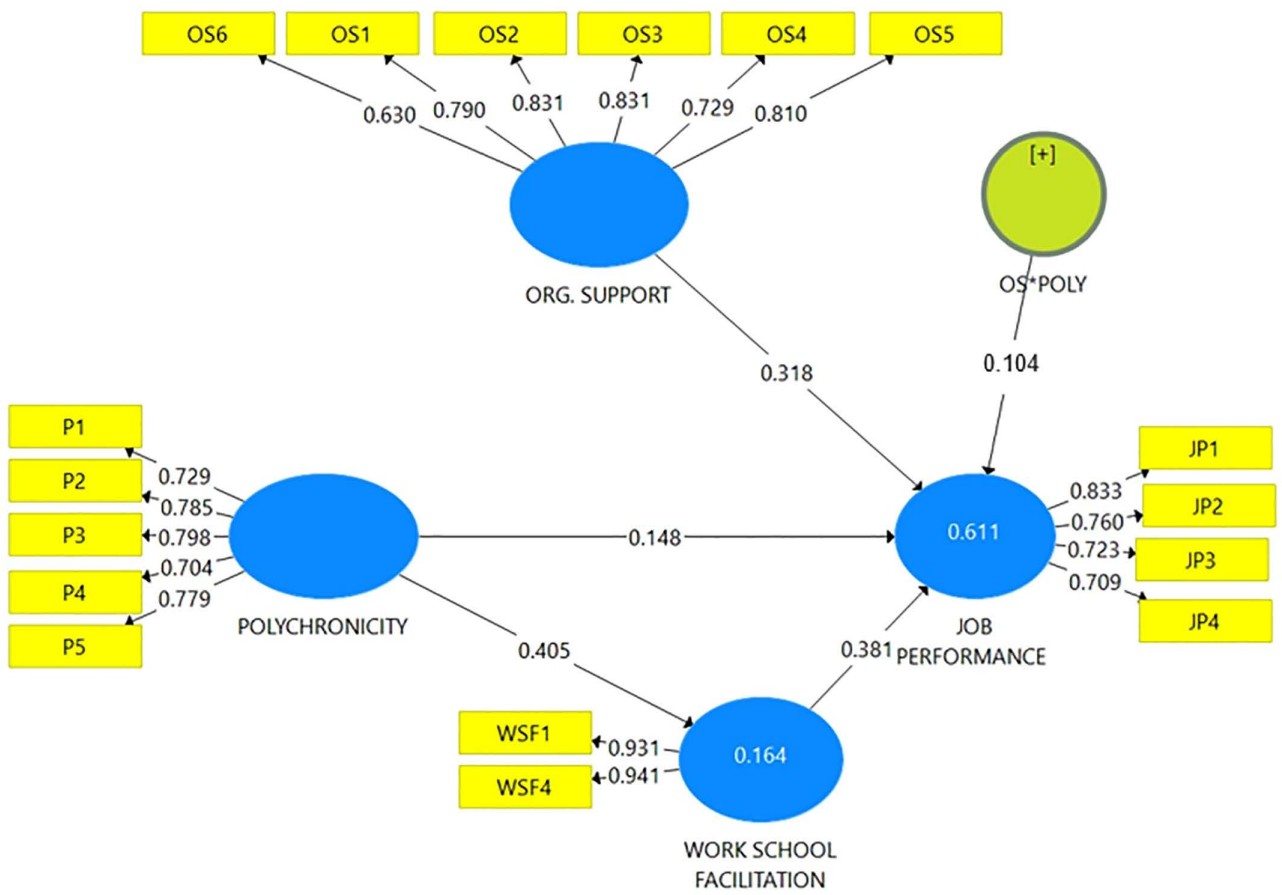

**Fig 2. Measurement Model.**

**Table 3. Construct Reliability and Validity.**

|  | Cronbach's Alpha | rho_A | Composite Reliability | (AVE) | McDonald Omega |
|---|---|---|---|---|---|
| Job Perf | 0.752 | 0.764 | 0.843 | 0.574 | 0.843 |
| Poly | 0.841 | 0.864 | 0.894 | 0.583 | 0.872 |
| Org. Support | 0.868 | 0.884 | 0.899 | 0.598 | 0.899 |
| WSF | 0.818 | 0.833 | 0.872 | 0.577 | 0.934 |

Note: Job Perf, job performance; Org. Support, organizational support; Poly, Polychronicity; WSF, Work School Facilitation.

Table 5 presented the VIF scores for the inner model of the constructs used in the model. Results from Table showed that all the VIF scores are less than 5 [67]. The VIF scores for the inner model therefore, showed that there is no threat of common method bias for all the constructs.

## Structural model

After adhering to the guidance provided by [55], our study proceeded by evaluating the statistical significance of the constructs used in the model to analyse hypotheses. This analysis focuses on scrutinizing the variance explained by the endogenous variables, as indicated by their $R^2$ values, as well as examining their path coefficients or standardized

 

**Table 4. Heterotrait-Montrait Ratio.**

| Discriminant Validity (DV) is frequently determined by the Heterotrait-Monotrait (HTMT) ratio because of its robustness and dependability [65]. The Heterotrait-Monotrait (HTMT) results for this study are presented in Table 4.Construct | Job Perf | ORG. SUPPORT | OS*POLY*JP | POLY | WSF |
|---|---|---|---|---|---|
| Job Perf | | | | | |
| ORG. SUPPORT | 0.721 | | | | |
| OS*POLY*JP | 0.534 | 0.210 | | | |
| POLY | 0.583 | 0.299 | 0.530 | | |
| WSF | 0.856 | 0.603 | 0.429 | 0.469 | |

Note: Job Perf = job performance, Org. Support (OS) = organizational support, Poly = Polychronicity, WSF = Work School Facilitation

**Table 5. Inner VIF values.**

| Construct | Job Perf |
|---|---|
| ORG. SUPPORT | 1.607 |
| POLY | 1.413 |
| WSF | 1.854 |

Note: Job Perf = job performance, Org. Support (OS) = organizational support, Poly = Polychronicity, WSF = Work School Facilitation

regression weights (Beta), and assessing their significance levels and effect size ($F^2$). To determine the statistical significance of the path coefficients, the researcher employed the bootstrapping approach. The findings are shown in Table 6, and 7 respectively. Again, the structural model is presented pictorially in Fig 3.

The results in Table 6 showed that polychronicity, organizational support, and work-school facilitation accounted for a strong variance in job performance (R-squared = 0.611) when all other factors not captured in this study but are affecting the job performance of postgraduate students are statistically controlled for. Thus, a 61.1% variance in the job performance of postgraduate students can be attributed to changes in polychronicity, organizational support, and work-school facilitation and its interaction effect. Again, Table 6 showed that polychronicity and organizational support accounted for a strong variance in work-school facilitation (R-squared = 0.164) when all other factors not captured in this study but are affecting the work-school facilitation of postgraduate students are statistically controlled for. Thus, a 16.4% variance in the work-school facilitation of postgraduate students can be attributed to changes in polychronicity and organizational support and its interaction effect.

The first hypothesis examines the relationship between Organizational Support and Job Performance. The path coefficient for this relationship is 0.316, and it is highly significant (p-value = 0.000). This result suggests that organizational support has a strong and positive influence on job performance. As organizational support increases, so does job performance. The significance of this result highlights the critical role that organizational support plays in enhancing employee performance. The second hypothesis explores the relationship between Polychronicity and Job Performance. Here, the path coefficient is 0.148, with a p-value of 0.020, which is statistically significant

**Table 6. Coefficient of Determination.**

| Constructs | R Square | R Square Adjusted |
|---|---|---|
| Job Perf | 0.611 | 0.600 |
| WSF | 0.164 | 0.158 |

Note: Job Perf, job performance; WSF, work school facilitation.

**Table 7. Path Co-efficient and Effect Size.**

| Decision Hypotheses | Constructs | Beta | F-Squared | T-Statistics | P-Values | CIL 2.5% | CIU 95% |
|---|---|---|---|---|---|---|---|
| | ORG. SUPPORT -> JOB PERF | 0.316 | 0.162 | 4.456 | 0.000 | 0.177 | 0.455 |
| H1- supported | POLY -> JOB PERF | 0.148 | 0.040 | 2.302 | 0.020 | 0.022 | 0.274 |
| H2-supported | OS*POLY -> JOB PERF | 0.104 | 0.047 | 2.455 | 0.016 | 0.021 | 0.187 |
| H3- supported | POLY -> WSF | 0.405 | 0.196 | 4844 | 0.000 | 0.240 | 0.570 |
| H5- supported | POLY -> WSF-> JOB PERF | 0.154 | 0.049 | 3.634 | 0.000 | 0.071 | 0.237 |
| H4-supported | WSF-> JOB PERF | 0.381 | 0.153 | 5.417 | 0.000 | 0.244 | 0.518 |

Note: Job Perf, job performance; Org. Support (OS), organizational support; Poly, polychronicity; WSF, work school; CIU/P, confidence interval (lower and upper).

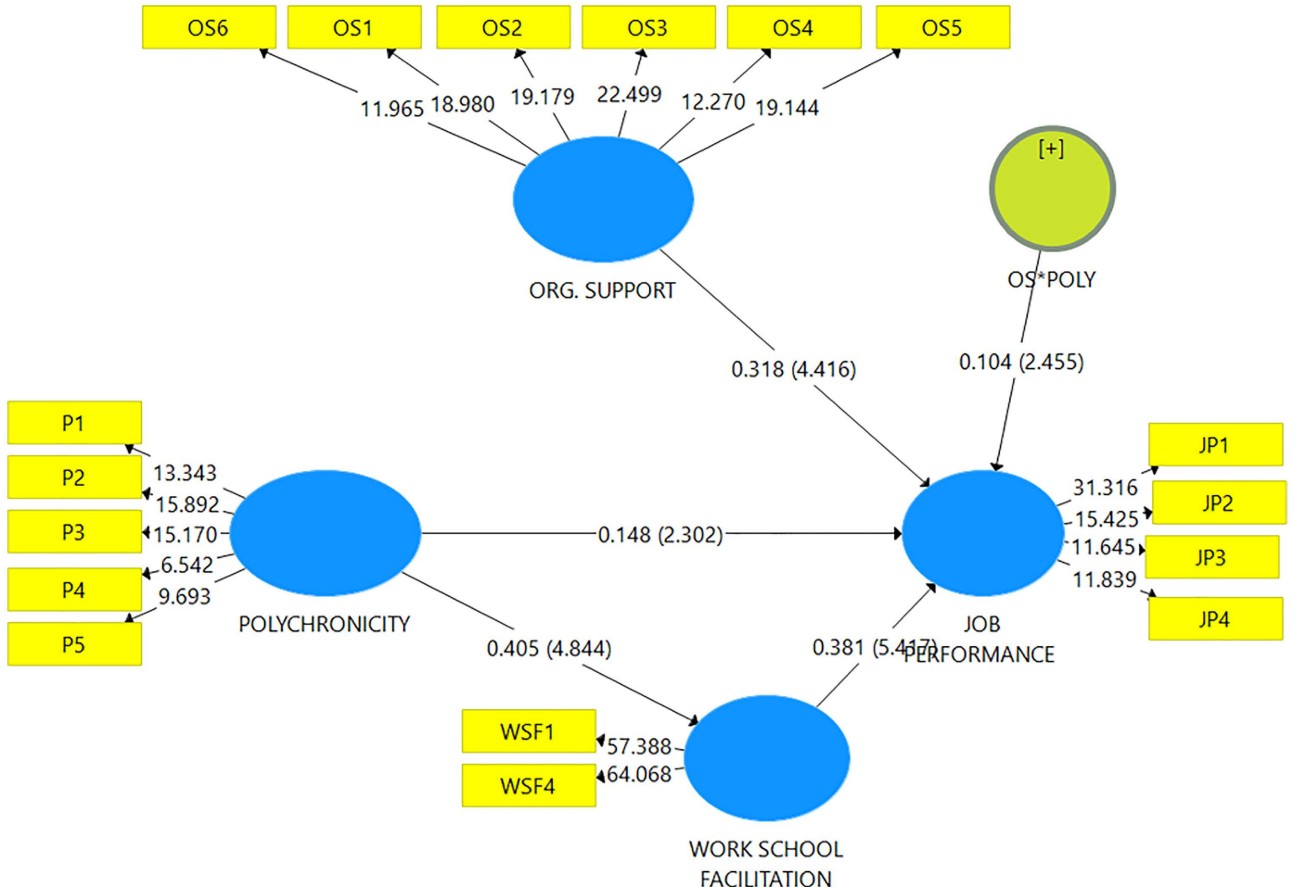

**Fig 3. Structural Model.**

at the 5% level. This indicates that polychronicity does have a positive effect on job performance, but the effect is smaller compared to other relationships in the model. While the relationship is significant, the relatively small beta coefficient suggests that changes in polychronicity alone are unlikely to lead to dramatic improvements in job performance.

**Table 8. Model fitness.**

| Fit Indices | Estimated Model |
|---|---|
| SRMR | 0.075 |
| Chi-Square | 0.062 |
| NFI | 0.930 |
| Rms Theta | 0.100 |

SRMR< 0.08; chi-square > 0.05; NFI > 0.90; rms < 0.12

The third hypothesis examines the interaction effect between Organizational Support and polychronicity on Job Performance. The interaction effect has a positive beta coefficient of 0.104, and the p-value is 0.016, indicating statistical significance. While the effect is positive, the magnitude of the interaction is modest compared to the individual effects of organizational support and polychronicity on job performance. This suggests that while the combination of organizational support and polychronicity can improve job performance, the interaction effect is not as powerful as the direct effects of these variables. It implies that the presence of both factors together contributes to job performance, but their combined impact is not as pronounced as their individual contributions. The fourth hypothesis tests the relationship between Work school facilitation and Job Performance. The path coefficient here is 0.381, and the result is highly significant with a p-value of 0.000. This represents the largest effect size in the model, indicating that workforce factors have the most substantial impact on job performance. This suggests that organizations aiming to enhance job performance should prioritize improving these Work-school facilitation, as they have the strongest influence on performance outcomes.

Finally, the fifth hypothesis looks at the indirect effect of Polychronicity on Job Performance through Workforce Factors. The beta coefficient for this indirect path is 0.154, with a p-value of 0.000, indicating statistical significance. While the indirect effect is positive and significant, the path coefficient is smaller compared to other relationships in the model, such as work-school facilitation directly influencing job performance. When considering the results in terms of their practical implications, the largest effect is the relationship between Work-school facilitation and Job Performance, with a beta coefficient of 0.381. This finding has significant consequences for organizations, as it suggests that workforce factors are the most important driver of job performance in the model. In contrast, the smallest effect is observed in the relationship between polychronicity and Job Performance, with a beta coefficient of 0.148. Although this relationship is statistically significant, the small effect size implies that policy changes alone are unlikely to result in substantial improvements in job performance. Evidence of the path coefficients and its T-statics are shown in Fig 2 and Fig 3 respectively.

### Model fit summary

It has been identified that all models must achieve a model fit [65,68]. For this reason, our study tested the fitness of the model used in this analysis of hypotheses. It is suggested by [69] and [70] that ($\chi^2$) an acceptable level is greater than 0.05. In the current study, it is obtained as 0.062. NFI is considered to be great with a value of 0.95 or above [71]. Also, [72] suggested that a NFI value of 0.90 or higher is an acceptable and good fit. Again, [71,72] posit that an acceptable SRMR value for a fit model should be less than 0.08. Finally, 0.12 or less is recommended for RMS theta [72]. Results in Table 8 showed that our model achieved fitness. This is because all fit indices achieved acceptable thresholds.

### Slope analysis

From Fig 4 it is seen that for employees with low organizational support, job performance remains relatively low and stable regardless of their level of polychronicity. This suggests that without adequate support from the organization, employees' ability to multitask or manage multiple tasks simultaneously (polychronicity) does not significantly enhance their job performance. In contrast, employees with high organizational support show a marked increase in job performance as their

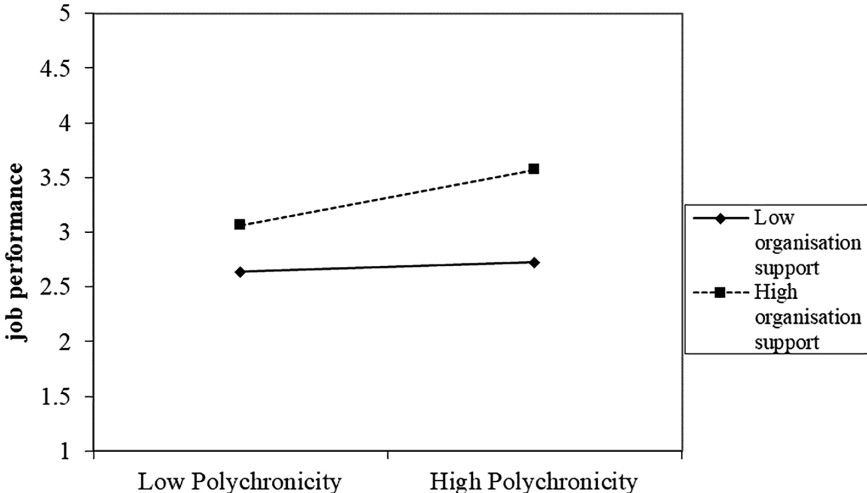

**Fig 4. Organizational support as a moderator between polychronicity and job performance.**

level of polychronicity increases. This indicates that when employees feel supported by their organization, their capacity to handle multiple tasks effectively translates into better job performance.

## Discussion

The analysis revealed that polychronicity significantly and positively affects job performance among Ghanaian postgraduate students (B = 0.149, p = 0.023). This suggests that individuals who can manage multiple tasks concurrently tend to perform better in both academic and work environments. These findings are consistent with earlier studies such as those by [11,12], and [36], which highlighted the advantages of multitasking for productivity and adaptability. The Ghanaian context may offer unique cultural strengths that support this outcome, contrasting with other sub-Saharan settings where multitasking is sometimes hindered by sociocultural factors [6]. Drawing on the Conservation of Resources theory [47], this result shows that individuals with multitasking skills are likely to perform better because they can effectively utilize and protect personal resources in dynamic settings. This accentuates the value of cultivating multitasking abilities as part of personal development programs within academic and professional institutions.

In practical terms, the finding points to the benefit of developing interventions aimed at enhancing multitasking skills among students and early-career professionals. Training modules, simulation-based learning, and collaborative academic tasks may help build these capabilities. It is important to note, however, that this finding diverges from research by [14] and [15], who concluded that multitasking contributes to performance only when adequate organizational support is present. This contrast invites further exploration into contextual and environmental conditions that may influence how polychronicity translates into actual performance outcomes.

The results also revealed that organizational support significantly strengthens the relationship between polychronicity and job performance (β = 0.103, p = 0.012), with a combined interaction effect of 0.252. This means that when students feel supported by their academic or professional institutions through mentorship, resource availability, or encouraging supervision, the benefits of multitasking are more fully realized. This finding aligns well with the Conservation of Resources theory, which emphasizes the role of supportive conditions in protecting and amplifying individual capabilities. The finding reinforces the importance of building enabling environments where multitasking talents can flourish. Institutions that provide timely feedback, adequate resources, and flexible work conditions may allow multitasking individuals to contribute more effectively. The result resonates with earlier findings by [14] and

[15], who found that in the absence of such support, multitasking may lead to stress or lower effectiveness. Therefore, support systems are not merely complementary but are central to optimizing the contribution of polychronic individuals.

Another key result from the study was the significant and positive relationship between polychronicity and work school facilitation (B = 0.159, p < 0.05). Students who are naturally inclined to juggle multiple responsibilities tend to integrate their academic and work roles more effectively. This finding is consistent with the conclusions of [39] and [59], who argued that multitasking helps individuals handle uncertainty and frequent transitions in modern work settings. This pattern highlights the utility of multitasking as a coping mechanism for managing dual-role demands. From a policy standpoint, this result suggests that higher education institutions and workplaces should consider creating programs that support students' ability to coordinate academic and professional responsibilities. Time management workshops, academic advising that considers employment schedules, and employer-sponsored learning initiatives may all contribute to better outcomes for individuals with multitasking strengths.

The results also showed a significant and positive effect of work-school facilitation on job performance (β = 0.381, p < 0.05), with strong statistical backing (t = 5.417, p = 0.000). This indicates that students who manage to balance school and work effectively tend to excel in their jobs. The finding lends further support to the Conservation of Resources theory, which posits that those who efficiently manage their personal resources, such as time and mental energy, are better equipped to meet performance expectations. This finding highlights the value of institutional flexibility. When schools and workplaces offer adaptable structures, such as variable class times or workload accommodations, individuals are more likely to manage stress and enhance job outcomes. For practitioners, the result highlights the need to consider the lived realities of postgraduate students when setting academic or work expectations.

The mediation analysis further revealed that work-school facilitation partially mediates the relationship between polychronicity and job performance (β = 0.055, p = 0.027). While multitasking directly contributes to improved performance, part of this effect is transmitted through the ability to balance school and work responsibilities. This adds a layer of complexity to our understanding of how multitasking operates in real-life scenarios. Rather than viewing polychronicity as a standalone trait, the findings suggest that its impact is contingent on how well individuals integrate multiple roles. According to the Conservation of Resources theory, this ability to manage dual responsibilities represents a valuable resource that enhances performance when supported effectively. Previous work by [25,27,37], and [38] echoes this view, stressing that individuals who experience work-school facilitation are more likely to apply their multitasking skills efficiently. This promotes resource conservation and reduces the potential strain associated with role conflict.

## Conclusion

This study sheds light on the significant role of polychronicity in enhancing job performance and work-school facilitation among postgraduate students in Ghana. The findings revealed that polychronic individuals, adept at managing multiple responsibilities, achieve better performance outcomes, especially in supportive organizational environments. Notably, organizational support amplifies the positive effects of polychronicity on job performance, underscoring the importance of resource-rich settings. Furthermore, work-school facilitation partially mediates the relationship between polychronicity and job performance, illustrating the critical role of balancing dual roles in maximizing multitasking potential. The study contributes to the Conservation of Resources (COR) theory by highlighting polychronicity as a valuable personal resource that interacts with organizational support to enhance performance. Additionally, it extends the theory by identifying work-school facilitation as a mediating resource that conserves time and energy, enabling efficient role management. This novel insight bridges gaps in literature by contextualizing polychronicity within the unique experiences of postgraduate students in a developing country setting.

## Implications of the study

The practical implications of the findings suggest that organizations should actively seek to identify and support employees with high polychronicity. By providing resources, training, and flexible work environments, companies can enhance the productivity and efficiency of postgraduate students, who are capable of handling multiple tasks simultaneously. This support is crucial in dynamic and fast-paced work environments, where the ability to switch between tasks seamlessly is highly valued. From a managerial perspective, understanding the strengths of polychronic individuals can inform better team management and task allocation. Managers should optimize team performance by assigning multitasking individuals to roles that require frequent task switching and adaptability.

Additionally, fostering a supportive work culture that emphasizes flexibility and resource availability can significantly boost overall job performance and employee satisfaction. Again, organizations should create policies and practices that enhance work-school facilitation. By supporting employees in balancing their educational and work commitments, organizations can indirectly boost job performance. This could involve offering flexible working hours, providing educational resources, or creating a supportive culture that values continuous learning and professional development.

Theoretically, the findings align with the Conservation of Resources (COR) theory, which posits that individuals strive to acquire and protect valuable resources. Polychronicity, as a resource, helps individuals manage their time, energy, and attention more effectively. The study extends COR theory by illustrating that organizational support acts as a crucial resource that enhances the positive impact of polychronicity on job performance. This emphasizes the importance of resource-rich environments in enabling employees to maximize their multitasking abilities, reducing stress, and improving performance outcomes.

## Limitation and future research

This study is limited by its focus on postgraduate students within the Ghanaian setting, which may not capture the cultural nuances or diverse experiences of students in other regions. Future studies could address this by exploring these relationships in diverse cultural settings to provide comparative insights and identify universal and context-specific factors. Additionally, the reliance on self-reported data, which could introduce bias, suggests that future research should incorporate objective measures or triangulate self-reports with observational or third-party evaluations to enhance validity. Furthermore, as this study did not account for the role of technological advancements like online learning platforms and productivity tools, future research could investigate how these technologies influence postgraduate students' ability to balance work, school, and other responsibilities, shedding light on innovative strategies to improve their academic and professional performance.

## Supporting information

**S1 Data. DATA for article.**
(CSV)

**S1 Appendix. Appendix A.**
(DOCX)

## Author contributions

**Conceptualization:** Salomey Ofori Appiah.

**Formal analysis:** Richard Kofi Boateng.

**Investigation:** Salomey Ofori Appiah, Richard Kofi Boateng.

**Methodology:** Salomey Ofori Appiah, Richard Kofi Boateng.

**Software:** Richard Kofi Boateng.

**Validation:** Richard Kofi Boateng.

**Writing – original draft:** Richard Kofi Boateng.

**Writing – review & editing:** Salomey Ofori Appiah.

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
