## [Decision Letter · Decision Letter 0]

2 Dec 2024

PONE-D-24-21809OPTIMISING POSTGRADUATE STUDENTS' JOB PERFORMANCE: THE IMPACT OF POLYCHRONICITY, WORK SCHOOL FACILITATION AND ORGANISATIONAL SUPPORTPLOS ONE

Dear Dr. Boateng,

Thank you for submitting your manuscript to PLOS ONE. After careful consideration, we feel that it has merit but does not fully meet PLOS ONE’s publication criteria as it currently stands. Therefore, we invite you to submit a revised version of the manuscript that addresses the points raised during the review process. Your manuscript has been evaluated by two reviewers, and their comments are appended below and in the attached files. The reviewers have provided a range of comments regarding various aspects of your manuscript, and particularly regarding the discussion section, and the omission of discussion of limitations to this study. Please ensure you address each of the reviewers' comments when revising your manuscript. Regarding the editorial note below, please also ensure you include with your revised manuscript files a letter from your Institutional Review Board (or equivalent research ethics committee) confirming that ethics approval was not required for this study. Please note that we may reject your manuscript if you do not provide this document as requested with your revised manuscript files. Finally, we note that one or more reviewers has recommended that you cite specific previously published works. As always, we recommend that you please review and evaluate the requested works to determine whether they are relevant and should be cited. It is not a requirement to cite these works. We appreciate your attention to this request.

We look forward to receiving your revised manuscript.

Kind regards,

Hugh Cowley

Staff Editor

PLOS ONE

Journal Requirements:

2. You indicated that ethical approval was not necessary for your study. We understand that the framework for ethical oversight requirements for studies of this type may differ depending on the setting and we would appreciate some further clarification regarding your research. Could you please provide further details on why your study is exempt from the need for approval and confirmation from your institutional review board or research ethics committee (e.g., in the form of a letter or email correspondence) that ethics review was not necessary for this study? Please include a copy of the correspondence as an "Other" file.

3. In the online submission form, you indicated that “The data underlying the results presented in the study are available from the corresponding author upon reasonable request.”. All PLOS journals now require all data underlying the findings described in their manuscript to be freely available to other researchers, either 1. In a public repository, 2. Within the manuscript itself, or 3. Uploaded as supplementary information. This policy applies to all data except where public deposition would breach compliance with the protocol approved by your research ethics board. If your data cannot be made publicly available for ethical or legal reasons (e.g., public availability would compromise patient privacy), please explain your reasons on resubmission and your exemption request will be escalated for approval.

4. Please ensure that you include a title page within your main document. You should list all authors and all affiliations as per our author instructions and clearly indicate the corresponding author.

5. Please ensure that you refer to Figure 1 in your text as, if accepted, production will need this reference to link the reader to the figure.

6. Please upload a copy of Figure 4, to which you refer in your text on page 19. If the figure is no longer to be included as part of the submission please remove all reference to it within the text.

8. Please remove your figures from within your manuscript file, leaving only the individual TIFF/EPS image files, uploaded separately. These will be automatically included in the reviewers’ PDF**.**

Reviewers' comments:

Reviewer's Responses to Questions

**Comments to the Author**

1. Is the manuscript technically sound, and do the data support the conclusions?

Reviewer #1: Yes

Reviewer #2: Yes

2. Has the statistical analysis been performed appropriately and rigorously? 

Reviewer #1: Yes

Reviewer #2: Yes

3. Have the authors made all data underlying the findings in their manuscript fully available?

Reviewer #1: Yes

Reviewer #2: Yes

4. Is the manuscript presented in an intelligible fashion and written in standard English?

Reviewer #1: Yes

Reviewer #2: Yes

5. Review Comments to the Author

Reviewer #1: The manuscript came with an attractive topic, a good and logical model, and the study is complete in its elements and appropriately coherent. There is a set of suggestions mentioned in the attached report.

Reviewer #2: REVIEW MANUSCRIPT PONE-D-24-21809

OPTIMIZING GHANAIAN POSTGRADUATE STUDENTS’ JOB PERFORMANCE:

THE IMPACT OF POLYCHRONICITY, WORK-SCHOOL FACILITATION, AND

ORGANISATIONAL SUPPORT

Dear Authors,

I have the opportunity to review this interesting article. Apart from its many advantages, several things need to be enriched and added to make it more perfect as a scientific work, among others:

Introduction

Add research gaps between variables.

Literature Review

• Add sufficient narrative or explanation of the conceptual framework.

• According to the conceptual framework, add a hypothesis about the relationship between Work School Facilitation and job performance.

Methodology

Add references and test for common method bias (CMB) adequately. As a guidances/references, see the link: https://doi.org/10.3390/educsci14040384;
https://doi.org/10.3390/jintelligence10030044.

Results

• Add a Results section with the order of presentation: Description and Correlation Analysis (Mean, Standard Deviation, and Correlation Coefficient of Variables); Measurement Model; Structural model; Fit Model; and Slope Analysis.

• Hypothesis test results should be interpreted adequately, including the addition of new hypotheses (Work School Facilitation with Job Performance).

Discussion of Hypotheses

• This section should be changed to just “Discussion.”

• Add an analysis of the relationship between Work School Facilitation and job performance.

• The discussion for each hypothesis should be sharpened and analyzed more critically, compared with previous research and research gaps.

Suggestions for future studies

In this section, change it to "Limitation and Future Research." Add limitations to this study and relate them to Future Research.

Conclusion

Add a conclusion section containing the main research findings. Show originality as a novelty. Complete the theoretical contribution and novelty of the findings.

6. PLOS authors have the option to publish the peer review history of their article (what does this mean? ). If published, this will include your full peer review and any attached files.

**Do you want your identity to be public for this peer review?** For information about this choice, including consent withdrawal, please see our Privacy Policy .

Reviewer #1: No

Reviewer #2: No

---

## [Author Response · Author response to Decision Letter 1]

20 Dec 2024

Thank you to all reviewers. we are pleased to informed you that your comments have been attended to

---

## [Decision Letter · Decision Letter 1]

11 Feb 2025

PONE-D-24-21809R1OPTIMISING POSTGRADUATE STUDENTS' JOB PERFORMANCE: THE IMPACT OF POLYCHRONICITY, WORK SCHOOL FACILITATION AND ORGANISATIONAL SUPPORTPLOS ONE

Dear Dr. Boateng,

Thank you for submitting your manuscript to PLOS ONE. After careful consideration, we feel that it has merit but does not fully meet PLOS ONE’s publication criteria as it currently stands. Therefore, we invite you to submit a revised version of the manuscript that addresses the points raised during the review process.

We look forward to receiving your revised manuscript.

Kind regards,

Hala Abdelgaffar, Ph.D.

Academic Editor

PLOS ONE

Additional Editor Comments:

Several key issues impact the validity, reliability, and results of this study. The use of convenience sampling introduces bias, which may limit the generalizability of the findings and affecting external validity. Please justify the use of convenience sampling and highlight practical constraints. Additionally, the sample size of 341 is based on Krejcie and Morgan’s formula, but the lack of a clear justification for its appropriateness raises concerns about reliability. Please justify the appropriateness of the sample size by explaining why the use of Krejcie and Morgan's formula is suitable for this particular study. The Harman’s single-factor test reveals common method bias with a variance of 44.13%, which is below the recommended threshold, potentially distorting relationships between variables and affecting internal validity. What measures did you use to reduce such bias. Finally, retaining weak indicators with factor loadings below 0.4 compromises construct validity and reliability. Please explain why specific indicators (e.g., OS6) were kept despite lower factor loadings.

Reviewers' comments:

Reviewer's Responses to Questions

**Comments to the Author**

1. If the authors have adequately addressed your comments raised in a previous round of review and you feel that this manuscript is now acceptable for publication, you may indicate that here to bypass the “Comments to the Author” section, enter your conflict of interest statement in the “Confidential to Editor” section, and submit your "Accept" recommendation.

Reviewer #1: All comments have been addressed

Reviewer #2: (No Response)

2. Is the manuscript technically sound, and do the data support the conclusions?

Reviewer #1: Yes

Reviewer #2: Yes

3. Has the statistical analysis been performed appropriately and rigorously? 

Reviewer #1: Yes

Reviewer #2: Yes

4. Have the authors made all data underlying the findings in their manuscript fully available?

Reviewer #1: Yes

Reviewer #2: Yes

5. Is the manuscript presented in an intelligible fashion and written in standard English?

Reviewer #1: Yes

Reviewer #2: Yes

6. Review Comments to the Author

Reviewer #1: The amendments are correct and achieve the required, just review the reference list and write reference number 13 correctly and completely.

Reviewer #2: REVIEW MANUSCRIPT PONE-D-24-21809-R1

OPTIMIZING GHANAIAN POSTGRADUATE STUDENTS’ JOB PERFORMANCE:

THE IMPACT OF POLYCHRONICITY, WORK-SCHOOL FACILITATION, AND

ORGANISATIONAL SUPPORT

Dear Authors,

I really appreciate your hard work in fulfilling my comments. Many thanks for it. However, several parts of the article still need improvement. Some of them which are quite crucial are:

The description of CMB is in the Methodology and Results Section—merge them into one. The citation source regarding CMB that you added does not explain CMB at all, so it would be better if you use the article references with the following links: https://doi.org/10.3390/educsci14040384;
https://doi.org/10.3390/jintelligence10030044. In addition, it would also be better if you added a description of what CMB is, its urgency, and how to overcome it. Both articles can be used as references.

In Table 7, sort H1 – H5. Hypothesis test results should be interpreted adequately. What do these results mean? All path coefficient values are positive; what does that mean? Also, explain which result is the largest and what its consequences are. Then, it also shows the smallest result along with its consequences.

7. PLOS authors have the option to publish the peer review history of their article (what does this mean? ). If published, this will include your full peer review and any attached files.

**Do you want your identity to be public for this peer review?** For information about this choice, including consent withdrawal, please see our Privacy Policy .

Reviewer #1: No

Reviewer #2: No

---

## [Author Response · Author response to Decision Letter 2]

14 Feb 2025

Thank you for all comments raised. As it stands we have attended to all comments and have used a red colour to indicate changes made in the manuscript

---

## [Decision Letter · Decision Letter 2]

28 Apr 2025

PONE-D-24-21809R2OPTIMIZING GHANAIAN POSTGRADUATE STUDENTS’ JOB PERFORMANCE: THE IMPACT OF POLYCHRONICITY, WORK-SCHOOL FACILITATION AND ORGANISATIONAL SUPPORTPLOS ONE

Dear Dr. Boateng

Thank you for submitting your manuscript to PLOS ONE. After careful consideration, we feel that it has merit but does not fully meet PLOS ONE’s publication criteria as it currently stands. Therefore, we invite you to submit a revised version of the manuscript that addresses the points raised during the review process. Please submit your revised manuscript by Jun 12 2025 11:59PM. If you will need more time than this to complete your revisions, please reply to this message or contact the journal office at plosone@plos.org . Please include the following items when submitting your revised manuscript:

We look forward to receiving your revised manuscript.

Kind regards,

Hala Abdelgaffar, Ph.D.

Academic Editor

PLOS ONE

Journal Requirements:

Reviewers' comments:

Reviewer's Responses to Questions

**Comments to the Author**

1. If the authors have adequately addressed your comments raised in a previous round of review and you feel that this manuscript is now acceptable for publication, you may indicate that here to bypass the “Comments to the Author” section, enter your conflict of interest statement in the “Confidential to Editor” section, and submit your "Accept" recommendation.

Reviewer #3: All comments have been addressed

2. Is the manuscript technically sound, and do the data support the conclusions?

Reviewer #3: Yes

3. Has the statistical analysis been performed appropriately and rigorously? 

Reviewer #3: Yes

4. Have the authors made all data underlying the findings in their manuscript fully available?

Reviewer #3: Yes

5. Is the manuscript presented in an intelligible fashion and written in standard English?

Reviewer #3: Yes

6. Review Comments to the Author

Reviewer #3: (No Response)

7. PLOS authors have the option to publish the peer review history of their article (what does this mean? ). If published, this will include your full peer review and any attached files.

**Do you want your identity to be public for this peer review?** For information about this choice, including consent withdrawal, please see our Privacy Policy .

Reviewer #3: No

---

## [Author Response · Author response to Decision Letter 3]

28 Apr 2025

Please review your reference list to ensure that it is complete and correct.

All references have been provided, and all omissions have been rectified.

For instance, all these references which were omitted have been provided in the revised manuscript.

1. Twaissi, N. M., Alhawtmeh, O. M., & O’la Hmoud Al-Laymoun. (2022). Polychronicity, job performance, and work engagement: The mediating role of supervisor’s organizational embodiment and moderation of psychological ownership. Cogent Business & Management, 9(1), 2143012.

2. Widodo, W., Gustari, I., & Chandrawaty, C. (2022). Adversity quotient promotes teachers’ professional competence more strongly than emotional intelligence: Evidence from Indonesia. Journal of Intelligence, 10(3), 44.

3. Damanik, J., & Widodo, W. (2024). Unlocking Teacher Professional Performance: Exploring Teaching Creativity in Transmitting Digital Literacy, Grit, and Instructional Quality. Education Sciences, 14(4), 384.

4. Fuller, Christie M., Marcia J. Simmering, Guclu Atinc, Yasemin Atinc, and Barry J. Babin. "Common methods variance detection in business research." Journal of business research 69, no. 8 (2016): 3192-3198.

5. Tehseen, S., Ramayah, T., & Sajilan, S. (2017). Testing and controlling for common method variance: A review of available methods. Journal of management sciences, 4(2), 142-168.

6. Kock, N. (2020). Harman’s single factor test in PLS-SEM: Checking for common method bias. Data Analysis Perspectives Journal, 2(2), 1-6.

7. Spector, P. E., Rosen, C. C., Richardson, H. A., Williams, L. J., & Johnson, R. E. (2019). A new perspective on method variance: A measure-centric approach. Journal of Management, 45(3), 855-880.\

8. Singh, A. S., & Masuku, M. B. (2014). Sampling techniques & determination of sample size in applied statistics research: An overview. International Journal of Economics, Commerce and Management, 2(11), 1-22.

9. Hossan, D., Dato’Mansor, Z., & Jaharuddin, N. S. (2023). Research population and sampling in quantitative study. International Journal of Business and Technopreneurship (IJBT), 13(3), 209-222.

---

## [Decision Letter · Decision Letter 3]

19 May 2025

PONE-D-24-21809R3OPTIMIZING GHANAIAN POSTGRADUATE STUDENTS’ JOB PERFORMANCE: THE IMPACT OF POLYCHRONICITY, WORK-SCHOOL FACILITATION AND ORGANISATIONAL SUPPORTPLOS ONE

Dear Dr. Boateng,

Thank you for submitting your manuscript to PLOS ONE. After careful consideration, we feel that it has merit but does not fully meet PLOS ONE’s publication criteria as it currently stands. Therefore, we invite you to submit a revised version of the manuscript that addresses the points raised during the review process.

We look forward to receiving your revised manuscript.

Kind regards,

Ahmed Abdelwahab Ibrahim El-Sayed

Academic Editor

PLOS ONE

Journal Requirements:

Additional Editor Comments:

This manuscript requires extensive editing to improve the clarity, accuracy, and consistency of English language usage, grammar, and academic style. Additionally, the research gap and significance of the study are not clearly articulated, making it difficult to understand how this work advances existing knowledge. The discussion section also needs substantial revision to present a balanced and critical debate, integrating the study's findings with both supporting and contradictory evidence from the current literature. At present, the discussion lacks depth, critical reflection, and fails to provide meaningful insights into the implications of the results.

Reviewers' comments:

Reviewer's Responses to Questions

**Comments to the Author**

1. If the authors have adequately addressed your comments raised in a previous round of review and you feel that this manuscript is now acceptable for publication, you may indicate that here to bypass the “Comments to the Author” section, enter your conflict of interest statement in the “Confidential to Editor” section, and submit your "Accept" recommendation.

Reviewer #4: All comments have been addressed

Reviewer #5: All comments have been addressed

Reviewer #6: All comments have been addressed

2. Is the manuscript technically sound, and do the data support the conclusions?

Reviewer #4: Yes

Reviewer #5: Partly

Reviewer #6: Yes

3. Has the statistical analysis been performed appropriately and rigorously? 

Reviewer #4: Yes

Reviewer #5: N/A

Reviewer #6: Yes

4. Have the authors made all data underlying the findings in their manuscript fully available?

Reviewer #4: Yes

Reviewer #5: Yes

Reviewer #6: Yes

5. Is the manuscript presented in an intelligible fashion and written in standard English?

Reviewer #4: Yes

Reviewer #5: Yes

Reviewer #6: Yes

6. Review Comments to the Author

Reviewer #4: Dear author,

Thank you for the opportunity to review this manuscript. Below are the comments on my review:

1. The subject matter of the article is relevant to be published in the journal.

2. The title clearly reflects the subject matter of the article.

3. The abstract and keywords do not provide good information about the article. It is not clear what the main objective of the research is.

4. Check whether all authors cited in the article are in the manuscript's references, and vice versa.

5. The article has problems with clarity and coherence in its language, which need to be reviewed. The Introduction does not make it clear what the objective of the research is, and what the research question (or questions) to be answered in the study are.

6. There is articulation between the theme and the theoretical basis. There is a good dialogue with the scientific literature in the field of knowledge.

7. There is analysis of the data and coherence in the argument. The data were categorized, organized, and analyzed in light of the theory that underpinned the research.

8. The bibliography used in the article is adequate and up-to-date.

This is an interesting study. However, I hope that these comments will help improve your manuscript.

Reviewer #5: Corrections given by the referees for the research were made. However, there are some things that need to be done in order to provide a more solid foundation for the research.

1. For reliability, McDonald Omega can be given in addition to Cronbach Alpha.

2.Tolerance and CI can be given for the multicollinearity problem.

3. It would be nice if a few more values such as RMSEA, CFI are given as model fit indexes.

4. Calculating the effect size of the study would be good for understanding the results reached in the findings.

5.Information on sampling can be presented in more detail if available.

6. Limitations related to the sampling method should be mentioned.

Reviewer #6: The authors are commended for the significant improvements made to the manuscript, particularly in response to previous feedback concerning the introduction, conceptual framework narrative, methodological details (sample justification, measurement specifics, CMB analysis), and the robustness of the results presentation and discussion. The inclusion and testing of new hypotheses (H4 and H5) have notably enriched the study's scope.

This study investigates the impact of polychronicity, work-school facilitation, and organizational support on the job performance of Ghanaian postgraduate students, grounded in the Conservation of Resources (COR) theory. The quantitative, explanatory design, employing a three-wave survey with 341 distance education postgraduate students to minimize common method bias, and the use of Structural Equation Modeling (SEM) with SMART PLS, are appropriate methodological choices.

The manuscript presents original research supported by a technically sound research design. The sample size justification, based on Krejcie and Morgan (1970), the use of validated scales (with clearly stated sources and item counts), and the application of PLS-SEM are all appropriate. The data analysis, encompassing common method bias (CMB) checks via time-lagged data collection, Harman's single-factor test, and correlation matrix (all strengthened in Rev 3), measurement model validation (factor loadings, Cronbach's Alpha, CR, rho_A, AVE, HTMT – detailed in Tables 3 & 4), and structural model assessment (VIF, R-squared, path coefficients, T-statistics, P-values, F-squared, model fit indices – presented in Tables 5-8), has been conducted rigorously and is well-reported. The conclusions drawn are generally supported by the presented data, which is made available according to PLOS ONE policy.

The manuscript is well-organized and presented intelligibly in standard English. While generally clear, a final light proofread to address minor grammatical points or awkward phrasing would further enhance its readability prior to publication.

In conclusion, this manuscript describes a technically sound piece of research with relevant findings. Provided the authors can definitively address and clarify the ethics statement to meet PLOS ONE's requirements, the manuscript will be very close to being acceptable for publication.

7. PLOS authors have the option to publish the peer review history of their article (what does this mean? ). If published, this will include your full peer review and any attached files.

**Do you want your identity to be public for this peer review?** For information about this choice, including consent withdrawal, please see our Privacy Policy .

Reviewer #4: **Yes: ** Gustavo Cunha de Araujo

Reviewer #5: **Yes: ** Osman AKHAN

Reviewer #6: No

---

## [Author Response · Author response to Decision Letter 4]

24 May 2025

Response to the journal comments. Actions taken are written in red ink in the manuscript.

Please review your reference list to ensure that it is complete and correct.

All references have been provided, and all omissions have been rectified.

For instance, all these references that were omitted have been provided in the revised manuscript.

Comments Response Page

Reviewer 4

The abstract and keywords do not provide good information about the article. It is not clear what the main objective of the research is. The abstract and keywords section has been rewritten to reflect the study’s purpose and objectives See page 1

The article has problems with clarity and coherence in its language, which need to be reviewed. The Introduction does not make it clear what the objective of the research is, and what the research question (or questions) to be answered in the study are. The objectives of the study have been provided in the introduction See the last paragraph of the introduction section on page 4

Reviewer 5

For reliability, McDonald Omega can be given in addition to Cronbach's Alpha. The McDonald Omega reliability test has been conducted with the help of the indicator loadings for each construct. See page 16 and Table 3

Tolerance and CI can be given for the multicollinearity problem Tolerance = 1−R2 (for each predictor in the regression). Since the researchers used PLS-SEM and not all R2 values for predictors are reported here, we typically used VIF (Variance Inflation Factor). However, the CI values are reported in Table 7.

See page 19 and Table 7

It would be nice if a few more values such as RMSEA, CFI are given as model fit indexes Good suggestion, however, the researcher used PLS-SEM and not CB-SEM. Thus, the indexes provided are given in PLS-SEM See page 21

Calculating the effect size of the study would be good for understanding the results reached in the findings This has been provided See page 19 and Table 7

Limitations related to the sampling method should be mentioned. This has been provided Page 10 in the method section

Reviewer 6

final light proofread to address minor grammatical points or awkward phrasing would further enhance its readability before publication. This has been done

Editor

The research gap and significance of the study are not clearly articulated, making it difficult to understand how this work advances existing knowledge. The discussion section also needs substantial revision to present a balanced and critical debate, integrating the study's findings with both supporting and contradictory evidence from the current literature. At present, the discussion lacks depth, critical reflection, and fails to provide meaningful insights into the implications of the results. The gaps that the study intends to fill are summarized in the last paragraph of the introduction

Again, the entire discussion section has been improved See the last paragraph of the introduction section on page 4

See the discussion section on pages 22-24

---

## [Decision Letter · Decision Letter 4]

11 Jul 2025

OPTIMIZING GHANAIAN POSTGRADUATE STUDENTS’ JOB PERFORMANCE: THE IMPACT OF POLYCHRONICITY, WORK-SCHOOL FACILITATION AND ORGANISATIONAL SUPPORT

PONE-D-24-21809R4

Dear Author,

We’re pleased to inform you that your manuscript has been judged scientifically suitable for publication and will be formally accepted for publication once it meets all outstanding technical requirements.

Kind regards,

Ahmed Abdelwahab Ibrahim El-Sayed

Academic Editor

PLOS ONE

Additional Editor Comments (optional):

Dear Authors,

Thank you for your contribution

I can accept your paper in its current form for publication in PLOS ONE

Reviewers' comments:

Reviewer's Responses to Questions

**Comments to the Author**

1. If the authors have adequately addressed your comments raised in a previous round of review and you feel that this manuscript is now acceptable for publication, you may indicate that here to bypass the “Comments to the Author” section, enter your conflict of interest statement in the “Confidential to Editor” section, and submit your "Accept" recommendation.

Reviewer #5: All comments have been addressed

2. Is the manuscript technically sound, and do the data support the conclusions?

Reviewer #5: Yes

3. Has the statistical analysis been performed appropriately and rigorously? 

Reviewer #5: Yes

4. Have the authors made all data underlying the findings in their manuscript fully available?

Reviewer #5: Yes

5. Is the manuscript presented in an intelligible fashion and written in standard English?

Reviewer #5: Yes

6. Review Comments to the Author

Reviewer #5: (No Response)

7. PLOS authors have the option to publish the peer review history of their article (what does this mean? ). If published, this will include your full peer review and any attached files.

**Do you want your identity to be public for this peer review?** For information about this choice, including consent withdrawal, please see our Privacy Policy .

Reviewer #5: No

---

## [Editor Report · Acceptance letter]

PONE-D-24-21809R4

PLOS ONE

Dear Dr. Boateng,

I'm pleased to inform you that your manuscript has been deemed suitable for publication in PLOS ONE. Congratulations! Your manuscript is now being handed over to our production team.

Kind regards,

on behalf of

Dr. Ahmed Abdelwahab Ibrahim El-Sayed

Academic Editor

PLOS ONE